# Economic uncertainty of pandemic and international airlines behaviour

**Ismail O. Fasanya**[1]*, **Oluwasegun B. Adekoya**[2], **Johnson A Oliyide**[2]

**1** School of Economics and Finance, University of the Witwatersrand, Johannesburg, South Africa,
**2** Department of Economics, Federal University of Agriculture, Abeokuta, Ogun State, Nigeria

* ismail.fasanya@wits.ac.za

## Abstract

This paper examines the role of uncertainty due to infectious diseases in predicting twenty International airline stocks within a nonparametric causality-in-quantiles framework. We observe that: First, the BDS test shows that nonlinearity is very important when examining the causal relationship between EMV-ID and airline stock returns and its volatility. Second, the nonparametric quantiles-based causality test shows that airline stocks predictability driven by pandemic-based uncertainty is stronger mostly around the lower quantiles, with weak evidences in middle and higher quantiles. Relevant policy implications can be drawn from these findings.

**Data Availability Statement:** The airline stocks data employed in this study is owned by Refinitiv Eikon. The authors accessed the data with their institution's membership. The authors had no special access privileges to the data others would

## 1. Introduction

One of the industries most vulnerable to shocks especially those that appear to be global in nature is the airline, and the broader aviation industry. These shocks can result from many sources, including financial turmoil, political and human factors (such as wars, political instabilities, terrorist attacks, epidemics) and natural factors (such as earthquakes, air disasters, volcanoes), and they impose inevitable detrimental effects on the performance of the industry [1]. Oftentimes, these unpalatable events lead to the restriction of travel which is followed by a significant credit crunch. This is further felt on the operational efficiency of the firms in the industry, thus resulting into unfavourable movements in their stock prices. As regards this, many studies have examined how some of these events impact on the performance of airline stock indices (see [1–8]).

Presently, the global economy is experiencing a serious health pandemic which has resulted in the underperformance of many stock markets. The COVID-19 pandemic has been claimed to be the health crisis with the greatest adverse impact on the performances of the global stock markets [9]. The pandemic has been demoralising to the entire airline industry [10] following the stay-at-home orders, general restrictions mandate, lock down policy and other policy interventions introduced by the governments of various countries to curb its spread. [11] reveals how the capacity of the airline industry has decreased at major carriers by about 60–80%. Around March when the effect of the pandemic gathered momentum, consumption was expected to decline following about 10 million claims of joblessness [12]. In light of this, recent studies have focused on the impact of the pandemic on different issues relating to the airline,

not have. The data underlying the results presented in the study are available from Thompson Reuters DataStream Database (https://eikon. thomsonreuters.com/index.html). While the data on infectious diseases uncertainty is freely available on http://policyuncertainty.com/infectious_EMV. html.

**Funding:** The authors received no specific funding for this work.

**Competing interests:** The authors have declared that no competing interests exist.

and aviation industry in general. For instance, its impact on transport volume and freight capacity [13], different components of the aviation industry (such as business traveler, consumer behaviour and demand-side, supply side, and so on-[14], airline employment [15] and consumption and labour supply shocks [16] has been examined.

Despite this large body of consideration of the impact of COVID-19 on the airline industry, less is known about how the industry's stock returns are driven by uncertainty induced by the pandemic. Like many other adverse events, it has induced uncertainty [15] which can further cause emotional responses to investors in their investment decisions as business and investments confidence are being eroded. Given this, the objective of this study is to examine the impact of uncertainty in equity markets due to infectious diseases on airline stocks during the COVID-19 pandemic period. We give credence to studies including [17] which consider the predictability of the aggregate stock returns of countries most affected by COVID-19 with health news, and [16], which examines the potential effects of pandemic on the seven scenarios in China including equity risk premium. To the best of our knowledge, the impact of uncertainty due to the outbreak of infectious diseases (which COVID-19 is the most notable recently) on the airline stock markets performances remains highly understudied.

In light of the aforementioned, our study offers insights into two important questions: Firstly, does uncertainty from health pandemics matter in tracking airline stock returns? In another way, does the inclusion of the pandemic uncertainty index in the standard capital assets pricing model (CAPM) improve the accuracy of the predictability of airline stock returns? Secondly, does the pandemic uncertainty induce airline stock returns volatility? Thirdly, does the effect of the uncertainty index on both stock returns and its volatility vary in quantiles? In so doing, we employ the causality-in-quantiles test originally due proposed by [18] in order to capture inherent nonlinearities, structural breaks and regime switches in the series. Apart from financial series often known to exhibit these unfavourable statistical features, the COVID-19 pandemic has resulted into wild movements in the series and notable structural shifts (see [19–21]). This makes linear models, such as the Granger-causality test to be unfit for this study as it could lead to spurious results resulting from misspecification bias. To confirm the suitability of the nonlinear causality test, however, the BDS test is first employed.

Apart from the introduction, the remaining parts of this study are in three sections. The immediate section describes the data and develops the methodology, followed by the discussion of results in the following section. The last section concludes the study.

## 2. Methodology and data

### 2.1 Methodology

As developed by [22, 23], the capital asset pricing model (CAPM) is a framework that is essential for investors in analysing a portfolio of assets that would maximise profit and also minimise associated risks. Notwithstanding, the traditional CAPM is restricted to examining the assets performance of the assets through returns. However, due to the need to sufficiently capture the extent to which an uncertainty induces shock into the stock market, hence causing it to become more volatile, this study, in addition to the first moment, also accounts for the uncertainty-induced effect of the pandemic on airline stocks at the second moment. With this, our paper follows the [24] methodology which is an extension of [18, 25] nonlinear causality frameworks. As noted by [18], the variable $x_t$ (EMV-ID) does not cause $y_t$ (airline stock returns) in the $\sigma - quantile$ with respect to the lag-vector of $\{y_{t-1}, \ldots, y_{t-q}, x_{t-1}, x_{t-q}\}$ if

$$Q_\sigma(y_t|y_{t-1}, \ldots, y_{t-q}, x_{t-1}, \ldots, x_{t-q}) = Q_\sigma(y_t|y_{t-1}, \ldots, y_{t-q}) \tag{1}$$

While $x_t$ causes $y_t$ in the $\sigma th$ quantile with respect to $\{y_{t-1}, \ldots, y_{t-q}, x_{t-1}, x_{t-q}\}$ if

$$Q_\sigma(y_t|y_{t-1}, \ldots, y_{t-q}, x_{t-1}, x_{t-q}) \neq Q_\sigma(y_t|y_{t-1}, \ldots, y_{t-q}) \tag{2}$$

Definitively, $Q_\sigma(y_t|\cdot) = \sigma th$ quantile of $y_t$ depending on $t$ and $0 < \sigma < 1$. We denote $V_{t-1} \equiv (y_{t-1}, \ldots, y_{t-q})$, $U_{t-1} \equiv (x_{t-1}, \ldots, x_{t-q})$, and $W_t = (U_t, V_t)$; and $F_{y_t|W_{t-1}}(y_t|W_{t-1})$ and $F_{y_t|V_{t-1}}(y_t|V_{t-1})$ represents the conditional distribution of $y_t$ given $W_{t-1}$ and $V_{t-1}$, respectively. Also, $F_{y_t|V_{t-1}}(y_t|V_{t-1})$ is assumed to be absolutely continuous in $y_t$ for almost all $W_{t-1}$. If we proceed by denoting $Q_\sigma(W_{t-1}) \equiv Q_\sigma(y_t|W_{t-1})$ and $Q_\sigma(V_{t-1}) \equiv Q_\sigma(y_t|V_{t-1})$, then we have $F_{y_t|W_{t-1}}\{Q_\sigma(y_t|W_{t-1})\} = \sigma$ with a probability of one. The hypothesis to be tested based on the specified definitions in Eqs (1) and (2) are;

$$H_0 = P\{F_{y_t|W_{t-1}}\{Q_\sigma(y_t|W_{t-1})\} = \sigma) = 1, \tag{3}$$

$$H_1 = P\{F_{y_t|W_{t-1}}\{Q_\sigma(y_t|W_{t-1})) = \sigma\} < 1, \tag{4}$$

Following [18], the distance measure $J = \{\tau_t E(\tau_t|W_{t-1})f_W(W_{t-1})\}$, where $\tau_t$ and $f_z(W_{t-1})$ are the regression error and marginal density function of $Z_{t-1}$, respectively. The regression error in Eq (3) can only be true if and only if $E[1\{y_t \leq Q_\sigma(V_{t-1})|W_{t-1}\} = \sigma$ or, equivalently, $1\{y_t \leq Q_\sigma(V_{t-1})\} = \sigma + \tau_t$, where $1\{\cdot\}$ is the indicator function. Thus, Jeong et al. (2012) specify the distance measure, $G \geq 0$, as:

$$G = E\left[\{F_{y_t|W_{t-1}}\{Q_\sigma(y_t|W_{t-1})\} - \sigma\}^2 f_W(W_{t-1})\right] \tag{5}$$

We will have a situation where $G = 0$ if and only if the null in Eq (3) is true, while we will have $G > 0$ otherwise in Eq (4). To test for the J-statistic, feasible kernel-function of Eq (6) is used:

$$\hat{G}_T = \frac{1}{T(T-1)s^{2q}}\sum_{t=q+1}^{T} \sum_{r=q+1,r\neq t}^{T} K\left(\frac{W_{t-1} - Z_{s-1}}{s}\right)\hat{\tau}_t\hat{\tau}_s, \tag{6}$$

Where $K(\cdot)$ denotes the kernel function with bandwidth $s$. T, q, $\hat{\tau}_t$ is the sample size, lag-order and estimate of the regression error, respectively. The estimate of the regression error is computed as thus:

$$\hat{\tau}_t = 1\{y_t \leq \hat{Q}_\sigma(Y_{t-1})\} - \sigma \tag{7}$$

Also, we further use the nonparametric kernel method to estimate the $\sigma th$ conditional quantile of $y_t$ given $V_{t-1}$ as $\hat{Q}_\sigma(V_{t-1}) = \hat{F}_{y_t|V_{t-1}}^{-1}(\sigma|V_{t-1})$, where the Nadarya-Watson Kernel estimator is specified as follows

$$\hat{F}_{y_t|V_{t-1}}(y_t|V_{t-1}) = \frac{\sum_{r=q+1,r\neq t}^{T} N\left(\frac{V_{t-1}-V_{r-1}}{s}\right)1(y_r \leq y_t)}{\sum_{r=q+1,r\neq t}^{T} N\left(\frac{V_{t-1}-V_{r-1}}{s}\right)} \tag{8}$$

Where $N(\cdot)$ is the kernel function and $s$ is the bandwidth.

[24] extends the above frameworks to account for causality in higher order moment, such that

$$y_t = h(V_{t-1}) + \vartheta(U_{t-1})\tau_t, \tag{9}$$

Where $\tau_t$ is the white noise process and $h(\cdot)$ and $\theta(\cdot)$ equals the unknown functions that satisfy pertinent conditions for stationarity. Although, this specification allows no granger-type

causality testing from $U_{t-1}$ to $y_t$, however, it could detect the "predictive power" from $U_{t-1}$ to $y_t^2$ when $\theta(\cdot)$ is a general nonlinear function. Thus, the study re-formulate Eq (9) to account for the null and alternative hypothesis for causality in variance in Eqs (10) and (11), respectively.

$$H_0 = P\{F_{y_t^2|W_{t-1}}\{Q_\sigma(y_t|W_{t-1})\} = \sigma\} = 1, \qquad (10)$$

$$H_1 = P\{F_{y_t^2|W_{t-1}}\{Q_\sigma(y_t|W_{t-1})\} = \sigma\} < 1, \qquad (11)$$

The feasible test statistic for the testing of the null hypothesis in Eq (10) is obtained, and then replace $y_t$ in Eqs (6)–(8) with $y_t^2$ (that is, volatility). With the inclusion of [18] approach, the study overcomes the issue that causality in mean implies causality in variance. Specifically, the study interprets the causality in higher-order moments through the use of the following model:

$$y_t = h(U_{t-1}, V_{t-1}) + \tau_t, \qquad (12)$$

Thus, we specify the higher order quantile causality as

$$H_0 = P\{F_{y_t^k|W_{t-1}}\{Q_\sigma(y_t|W_{t-1})\} = \sigma\} = 1, \; for \; k = 1, 2, \ldots, k, \qquad (13)$$

$$H_1 = P\{F_{y_t^k|W_{t-1}}\{Q_\sigma(y_t|W_{t-1})\} = \sigma\} < 1, \; for \; k = 1, 2, \ldots, k. \qquad (14)$$

Overall, we test that $x_t$ granger causes $y_t$ in $\sigma th$ quantile up to the K-th moment through the use of Eq (13) to construct the test statistic of Eq (6) for each k. Although [25], note that it is not easy to combine different statistics for each $k = 1, 2, \ldots, k$ into one statistic for the joint null in Eq (13) which is mutually correlated. However, to circumvent this issue, we adopt a sequential-testing method as described by Nishiyama et al. (2011) with some modifications. To begin with, we test for the nonparametric granger causality in mean (k = 1). Failure to reject the null of k = 1 does not translate into non causality in variance, thus, we construct the tests for k = 2. Finally, we test for the existence of causality-in-mean and variance successively.

## 2.2 Data sources and description

This paper utilises daily data of 20 international airline stock prices and Infectious Diseases Uncertainty (EMV-ID) index covering the period from December 09, 2013 to July 31, 2020, 1735 daily groups based on data availability. The selected airlines are based on data availability and international influence with their two-letter designations from the International Air Transport Association (IATA). These include American Airlines, Air France-KLM, Ana Holdings, Anaam Intl. Holding Group, Asiana Airlines, Cathay Pacific Airways, China Eastern Airlines, China Airlines, China Southern, Delta Airlines, Deutsche Lufthansa, EVA Air, Hainan Airlines, Korean Airlines, Malaysian Airlines, Qantas Airways, Singapore Airlines, Southwest Airlines, Thai Airways and United Airlines, all sourced from DataStream database and EMV-ID data collected from http://policyuncertainty.com/infectious_EMV.html. Table 1 shows the descriptive analysis of daily returns and EMV-ID (We decided to work with logarithmic returns series of airline stock prices and levels of EMV-ID index for non-causality to hold).

For ensuring due diligence in most empirical studies, the prerequisite knowledge about the statistical properties of the underlying series be first provided. We keep this norm by revealing brief statistical information about the series. The descriptive statistics reported in Table 1 shows that on average, the highest return is observed for China South airline, Thai airline has the lowest return. Most of the airline stocks exhibit negative returns on average over the period

**Table 1. Preliminary analysis.**

| | Mean | Maximum | Minimum | Std. Dev. | Skewness | Kurtosis | Jarque-Bera | Q-stat | | NG-Perron | Dickey Fuller GLS |
|---|---|---|---|---|---|---|---|---|---|---|---|
| | | | | | | | | **4** | **8** | | |
| American | -0.046 | 34.428 | -29.068 | 3.106 | 0.613 | 24.732 | 34230.600*** | 18.866*** | 44.808*** | -847.901*** | -35.907*** |
| Ana | 0.002 | 10.087 | -10.503 | 1.529 | -0.048 | 10.478 | 4040.365*** | 18.414*** | 24.858*** | -866.387*** | -41.142*** |
| Anaam | -0.048 | 9.535 | -10.536 | 2.455 | -0.198 | 8.463 | 2167.176*** | 3.982 | 4.625 | -1036.600*** | -30.814*** |
| Asiana | -0.010 | 26.236 | -35.579 | 2.766 | 0.240 | 33.712 | 68165.790*** | 4.444 | 6.854 | -863.510*** | -39.240*** |
| Cathay | -0.057 | 7.483 | -7.994 | 1.559 | 0.047 | 5.543 | 467.837*** | 4.457 | 14.388* | -861.656*** | -38.613*** |
| China | -0.017 | 9.417 | -10.437 | 1.564 | -0.028 | 9.854 | 3394.122*** | 3.225 | 14.880* | -567.726*** | -19.997*** |
| China-east | 0.023 | 9.616 | -10.582 | 2.657 | -0.004 | 7.146 | 1241.640*** | 3.225 | 14.880* | -853.448*** | -36.786*** |
| China-south | 0.035 | 9.628 | -10.581 | 2.6963 | -0.035 | 6.850 | 1071.062*** | 2.889 | 21.740*** | -855.960*** | -37.254*** |
| Delta | -0.008 | 19.076 | -30.100 | 2.475 | -1.287 | 27.394 | 43472.990*** | 8.583** | 53.487*** | -587.010*** | -14.996*** |
| Deutsche | -0.042 | 10.314 | -14.934 | 2.161 | -0.631 | 8.696 | 2458.602*** | 5.051 | 12.340 | -716.805*** | -25.996*** |
| Eva | -0.013 | 8.359 | -10.471 | 1.555 | -0.051 | 10.490 | 4054.280*** | 2.370 | 8.002 | -861.176*** | -38.464*** |
| Hainan | -0.015 | 9.658 | -11.050 | 2.135 | -0.095 | 11.198 | 4858.085*** | 21.759*** | 51.745*** | -531.605*** | -18.458*** |
| France | -0.042 | 12.872 | -13.576 | 2.605 | -0.240 | 6.102 | 711.867*** | 3.175 | 12.562 | -857.889*** | -37.649****** |
| Malaysia | -0.053 | 26.826 | -19.284 | 2.669 | 0.487 | 16.903 | 14033.880*** | 5.786 | 11.272 | -866.373*** | -41.117*** |
| Qantas | 0.068 | 23.313 | -16.741 | 2.250 | 0.136 | 14.971 | 10359.420*** | 6.256 | 18.445** | -20.376** | -4.572*** |
| Shandong | -0.048 | 9.549 | -10.584 | 2.274 | -0.522 | 8.809 | 2516.708*** | 2.933 | 21.172*** | -844.226*** | -35.402****** |
| Singapore | -0.043 | 9.859 | -11.613 | 1.243 | -0.385 | 15.769 | 11823.700*** | 0.699 | 3.556 | -32.553*** | -5.807*** |
| Southwest | 0.030 | 13.492 | -16.381 | 2.142 | -0.515 | 11.926 | 5832.509*** | 7.219 | 17.079** | -866.471*** | -41.860*** |
| Thai | -0.099 | 18.430 | -16.179 | 2.941 | 0.565 | 10.876 | 4573.954*** | 11.923** | 26.163*** | -744.783*** | -26.584*** |
| United | -0.010 | 22.884 | -36.083 | 3.101 | -1.094 | 25.344 | 36416.620*** | 8.875* | 56.971*** | -1396.560*** | -16.232*** |
| EMV_ID | 2.296 | 68.370 | 0.000 | 7.657 | 4.757 | 28.034 | 51820.130*** | 152.450*** | 163.48*** | -32.839*** | -3.648862*** |

***,**,* confirms significance at 1%, 5% and 10% respectively.

under consideration, indicating a likely evidence of loss by investors in these industries and increasing function in terms of the disease rate. In terms of their volatilities, we observe that with the standard deviation value of EMV-ID which is 7,657, it is more than twice as volatile as the most volatile airlines, American and United, whose standard deviation statistics are 3.106 and 3.101 respectively, as these are evident in Fig 1. Singapore airline exhibits the least degree of variability. The implication of this is that uncertainty due to infectious diseases variability speaks volume in the direction and behaviour of investors and other market participants in risk hedging. It is not surprising how that the Jarque-Bera test rejects the null hypothesis of normal distribution for all the series following from the reports of both the skewness and kurtosis statistics. While the skewness values hover between positive and negative for all the returns series, their kurtosis estimates are exceedingly larger than the standard threshold. This suggests the presence of extreme fluctuations in airline industry.

Certain implications could be drawn from this brief descriptive analysis. Firstly, the non-normality of the series gives a relative indication of heavy right or left tail and excess kurtosis. This could further suggest the presence of nonlinearity and/or structural shifts along the time paths of the series such that the use of linear or constant parameter models would bring about spurious results. This gives a concrete justification for our choice of quantiles-based causality test. Secondly, the evidence of heavy tails as well as high volatility passes motivates the necessity to examine the relationship in both the conditional-mean and conditional-variance [24].

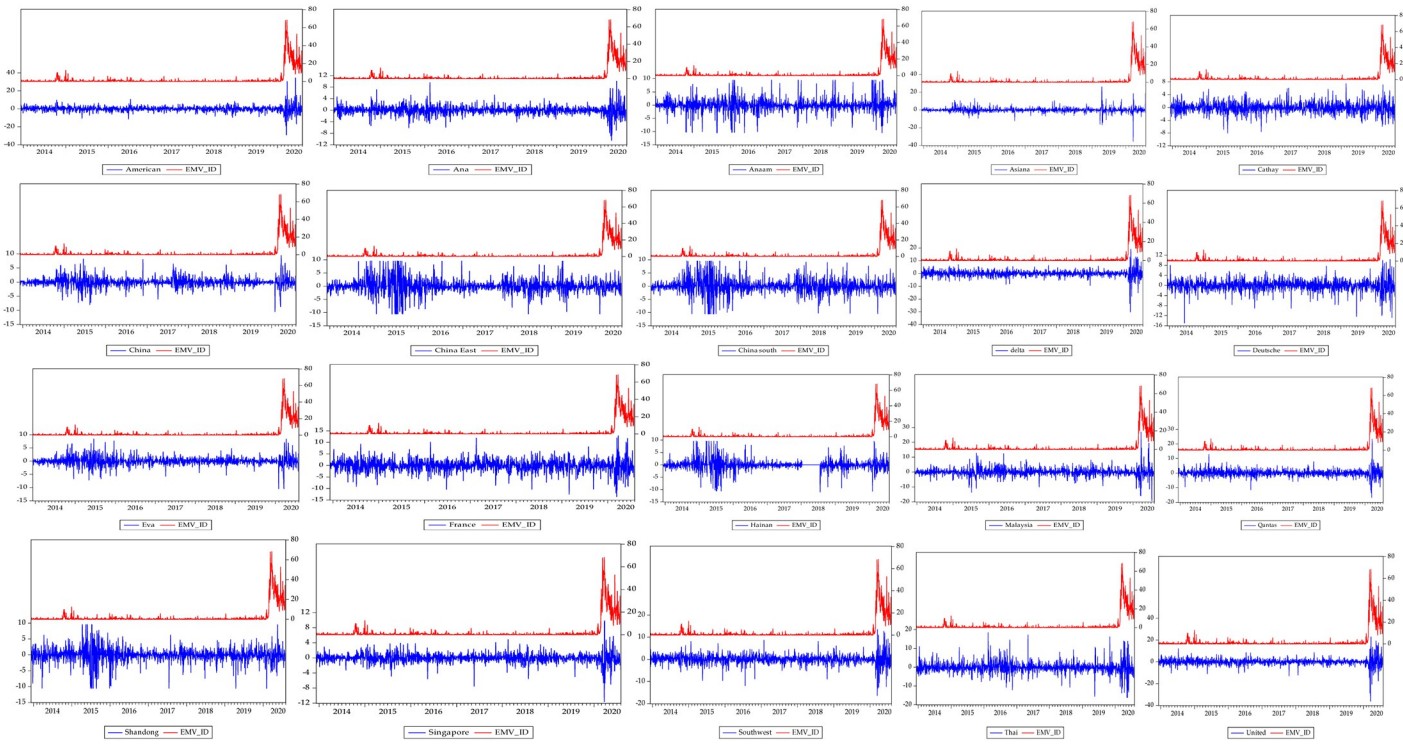

**Fig 1. Trends of the international airlines stock returns and equity market volatility infectious diseases.**

## 3. Discussion of empirical result

### 3.1 Linear causality

We present the linear estimation results between uncertainty due to infectious diseases and airline stocks, which aim to test for the predictability of airline stocks with EMV-ID for the full sample and during the Covid pandemic. The standard granger causality test entails a first approximation to the presence of causal relationship between the variables of interest, thus, we commence this section by examining the linear causal effect from EMV_ID to the returns of the airline stock. As shown in Table 2, barring American and Qantas airlines for full sample and Ana during Covid, there is no evidence of predictability originating from the EMV-ID for International airline stock returns at the conventional percent level of significance. Overall, the evidence is insignificant and weak in terms of the ability of disease-based uncertainty to predict airline stock returns. We suspect that the large acceptance of the non-causality hypothesis might be due to the non-linearity in the data structure. Thus, we motivate the suitability of the nonparametric non-linear causality method by testing for the presence of non-linearity among the variables of interest through the use of BDS test by [26] test.

As presented in Table 3, the BDS test suggest that utilising a linear model to capture the causal flows from EMV_ID to the airline stock returns might be inappropriate, hence, it is expedient to analyse the causal flows by utilising a technique that can incorporate the non-linear nature of the variables of interest.

### 3.2 Nonlinear causality

Owing to the question raised in the introduction section about the possibility of the effect of pandemic-induced uncertainty on airline stocks to vary across quantiles, this section is

**Table 2. Linear causality of [27].**

| EMV_ID does not cause | F-stat | | EMV_ID does not cause | F-stat | |
|---|---|---|---|---|---|
| | Full | During Covid | | Full | During Covid |
| AMERICAN | 3.170** | 1.133 | EVA | 0.751 | 0.296 |
| ANA | 0.248 | 3.351** | HAINAN | 0.490 | 0.912 |
| ANAAM | 0.988 | 1.639 | FRANCE | 0.156 | 0.033 |
| ASIANA | 0.336 | 1.012 | MALAYSIA | 1.030 | 0.214 |
| CATHAY | 0.956 | 0.216 | QANTAS | 4.796*** | 0.250 |
| CHINA | 0.514 | 0.173 | SHANDONG | 0.303 | 0.624 |
| CHINAEST | 0.389 | 0.008 | SINGAPORE | 1.062 | 0.351 |
| CHINASOU | 0.283 | 0.014 | SOUTHWEST | 2.286 | 1.432 |
| DELTA | 2.263 | 0.974 | THAI | 0.003 | 0.606 |
| DEUTSCHE | 0.067 | 0.205 | United | 2.239 | 0.807 |

***,**,* confirms significance at 1%, 5% and 10% respectively.

dedicated to delving into this research possibility through the employed non-linear causality in quantiles technique of [24]. It is important to document that the quantiles are partitioned into three, each representing different market conditions- that is, the lower quantiles (the bearish market condition), the middle quantiles (the normal market condition) and the higher quantiles (the bullish market condition). During bearish market condition, stocks experience market downturn which is mostly evident in a continuous fall in prices. As highlighted by

**Table 3. Brock et al. (1996)—(BDS) test.**

| | Dimensions | | | | |
|---|---|---|---|---|---|
| | 2 | 3 | 4 | 5 | 6 |
| AMERICAN | 0.023*** | 0.043*** | 0.053*** | 0.054*** | 0.053*** |
| ANA | 0.022*** | 0.045*** | 0.060*** | 0.066*** | 0.068*** |
| ANAAM | 0.039*** | 0.071*** | 0.090*** | 0.097*** | 0.099*** |
| ASIANA | 0.023*** | 0.043*** | 0.056*** | 0.061*** | 0.062*** |
| CATHAY | 0.011*** | 0.022*** | 0.027*** | 0.032*** | 0.034*** |
| CHINA | 0.029*** | 0.057*** | 0.075*** | 0.087*** | 0.095*** |
| CHINAEST | 0.037*** | 0.071*** | 0.093*** | 0.106*** | 0.110*** |
| CHINASOU | 0.033*** | 0.063*** | 0.081*** | 0.091*** | 0.095*** |
| DELTA | 0.024*** | 0.039*** | 0.050*** | 0.055*** | 0.055*** |
| DEUTSCHE | 0.013*** | 0.025*** | 0.031*** | 0.033*** | 0.032*** |
| EVA | 0.031*** | 0.059*** | 0.076*** | 0.087*** | 0.093*** |
| HAINAN | 0.053*** | 0.102*** | 0.136*** | 0.158*** | 0.168*** |
| FRANCE | 0.016*** | 0.026*** | 0.033*** | 0.036*** | 0.035*** |
| MALAYSIA | 0.033*** | 0.055*** | 0.068*** | 0.076*** | 0.080*** |
| QANTAS | 0.019*** | 0.036*** | 0.047*** | 0.051*** | 0.051*** |
| SHANDONG | 0.031*** | 0.057*** | 0.073*** | 0.080*** | 0.083*** |
| SINGAPORE | 0.019*** | 0.037*** | 0.047*** | 0.054*** | 0.057*** |
| SOUTHWEST | 0.024*** | 0.047*** | 0.062*** | 0.067*** | 0.067*** |
| THAI | 0.024*** | 0.050*** | 0.064*** | 0.073*** | 0.075*** |
| UNITED | 0.028*** | 0.051*** | 0.064*** | 0.070*** | 0.072*** |

***,**,* confirms significance at 1%, 5% and 10% respectively.

[28], such periods are usually caused by unfavourable market conditions, uncertainty resulting from pandemics, and sentiments towards future uncertainty in the market. Unlike the bearish market condition, the bullish market condition comes to play when markets are rising or are expected to rise in the future. This usually sponsors the motives of investors to commit more resources to such market. When a market is in a normal condition, the stock performance remains stable for a long time with low volatility and an absence of a large gap in prices. In respect of these conditions, uncertainty has the tendency to induce shocks in the market, hence causing volatility and market uncertainty.

In section 3.2.1, airline stock returns' predictability by pandemics' uncertainty is considered at different market conditions, while section 3.2.2 is dedicated to the predictability of the volatility of the stock indices.

**3.2.1. Does uncertainty in pandemic matter in the predictability of the airline stock returns?.** As noted earlier, we present the summary of the result of the nonparametric causality in mean in Table 4, which corroborates the findings presented in Fig 2. For the causality in mean for the full sample period as shown in Table 4, most airline stock returns significantly respond to the shock induced by health pandemics at bearish market conditions. In this case, uncertainty induced by health pandemics generally affects the stock returns of Ana, Delta, Malaysia, Qantas, Shandong, Singapore, Southwest, Thai and United when the market is trending unfavourably (bearish market condition). However, in the case of Cathay, China and Hainan, uncertainty induced by health pandemic affects their stock returns when the market is well-behaved (normal market condition).

The pandemic-induced uncertainty during the COVID-19 pandemic influenced fifteen of the airline stocks returns, which is higher than twelve recorded in the full sample period. In

**Table 4. Summary of causality in mean (that is, returns) for both full and during COVID-19 samples.**

|  |  | Full Sample | | | During COVID-19 | | |
|---|---|---|---|---|---|---|---|
|  | $H_0$: EMV_ID does not Granger-cause: | LQ | MQ | HQ | LQ | MQ | HQ |
| 1 | American | No | No | No | Yes | No | No |
| 2 | Ana | Yes | No | No | No | No | No |
| 3 | Anaam | No | No | No | Yes | No | No |
| 4 | Asiana | No | No | No | Yes | No | No |
| 5 | Cathay | No | Yes | No | Yes | No | No |
| 6 | China | No | Yes | No | No | Yes | No |
| 7 | China east | No | No | No | Yes | No | No |
| 8 | China south | No | No | No | Yes | No | No |
| 9 | Delta | Yes | No | No | No | No | No |
| 10 | Deutsche | No | No | No | Yes | No | No |
| 11 | Eva | No | No | No | Yes | No | No |
| 12 | France | No | No | No | No | No | No |
| 13 | Hainan | No | Yes | No | Yes | No | No |
| 14 | Malaysia | Yes | No | No | Yes | No | No |
| 15 | Qantas | Yes | No | No | Yes | No | No |
| 16 | Shandong | Yes | No | No | Yes | No | No |
| 17 | Singapore | Yes | No | No | Yes | No | No |
| 18 | Southwest | Yes | No | No | No | No | No |
| 19 | Thai | Yes | No | No | Yes | No | No |
| 20 | United | Yes | No | No | No | No | No |

Note: LQ represents lower quantiles (that is, quantiles 0.1 to 0.3), MQ is middle quantiles (that is, 0.4 to 0.6), and HQ is higher quantiles (that is, 0.7 to 0.9).

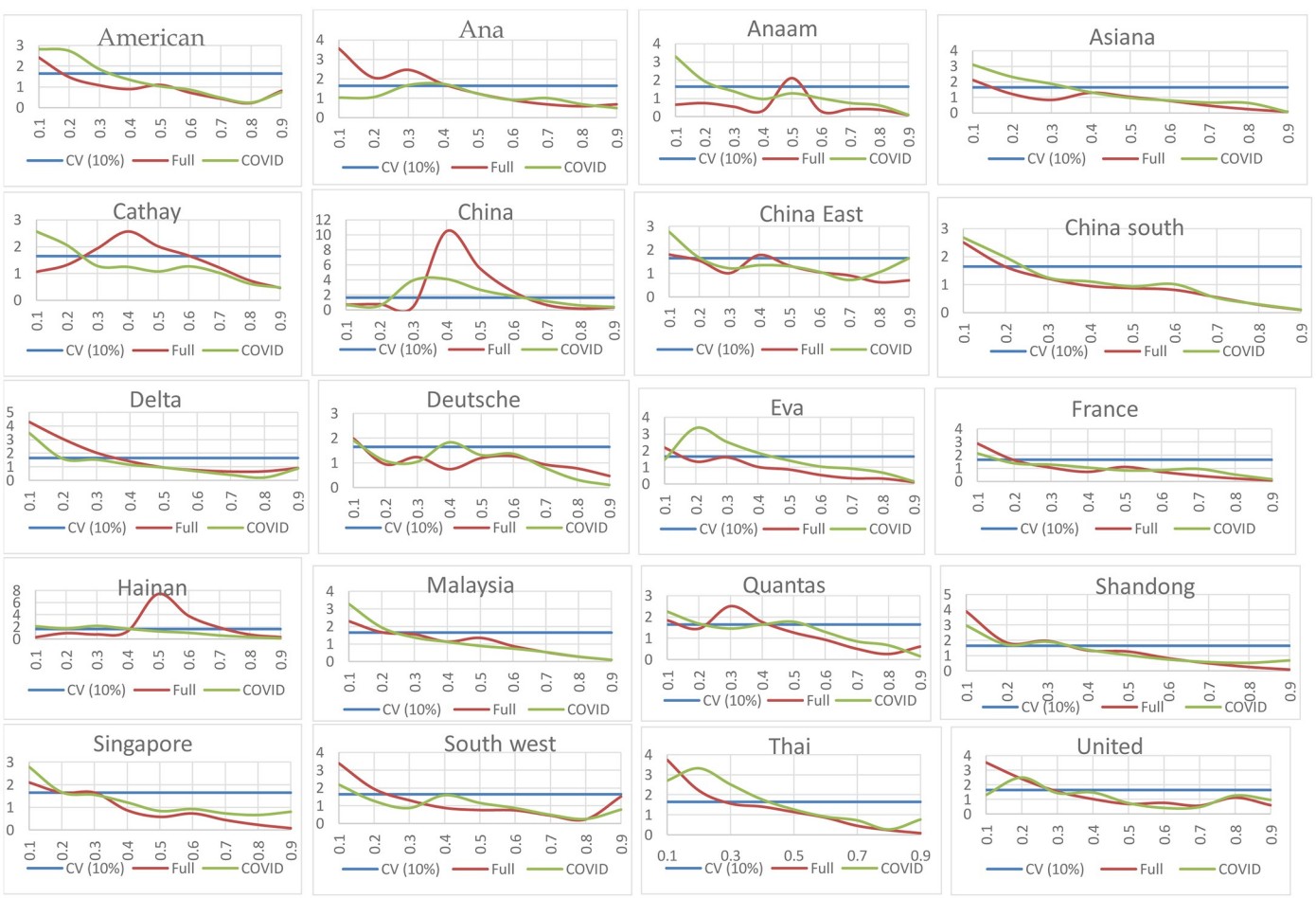

**Fig 2. Causality-in-mean result for the airlines.** Note: The figure plots the estimates of the nonparametric causality tests of the various quantiles. The y-axis report test statistics and quantiles of the airline stock returns are on x-axis. Full and COVID represents the full sample and COVID-19 pandemic sample, respectively.

detail, the stocks of American, Anaam, Asiana, Cathay, China east, china south, Deutche, Eva, Hainan, Malaysia, Qantas, Shandong, Singapore and Thai airlines were affected when the markets are poorly-coordinated during the COVID-19 pandemic. In contrast, only China airline stock returns is influenced when the market is behaving normally.

In summary, we observe a mixed responsiveness of the airlines stock returns to uncertainty induced by pandemic over the quantiles for both samples (that is, full sample and during covid-19 period). As can be seen from both periods, the null hypothesis of no Granger causality is overwhelmingly rejected in mostly at 10 percent level of significance at the lower quantiles and in few cases at the middle quantiles—a result which starkly contrasts the weak evidence of predictability under the linear framework. Intuitively, predictive power of uncertainty induced by the health pandemic is strongest around the lower quantiles and for few cases around the median.

**3.2.2 Does uncertainty in pandemic matter in the predictability of the airline stock volatility?.** Moving to the causality in variance as shown in Table 5 for the full sample and during COVID-19 period as evident in Fig 3. The causality in variance result reveals the susceptibility of the stock returns to uncertainty induced by health pandemics hence causing the stock market to be volatile. As characterized in Fig 3, for the full sample period, the causal flow from the

**Table 5. Summary of causality in variance (that is, volatility) for both full and during COVID-19 samples.**

| | | Full Sample | | | During COVID-19 | | |
|---|---|---|---|---|---|---|---|
| | $H_0$: EMV_ID does not Granger-cause: | LQ | MQ | HQ | LQ | MQ | HQ |
| 1 | American | Yes | No | No | Yes | No | No |
| 2 | Ana | Yes | No | No | No | No | No |
| 3 | Anaam | No | No | No | No | No | No |
| 4 | Asiana | No | No | No | Yes | No | No |
| 5 | Cathay | No | No | No | Yes | Yes | No |
| 6 | China | Yes | No | No | Yes | No | No |
| 7 | China east | No | No | No | Yes | No | No |
| 8 | China south | No | No | No | Yes | No | No |
| 9 | Delta | Yes | No | No | Yes | No | No |
| 10 | Deutsche | No | No | No | Yes | No | No |
| 11 | Eva | No | No | No | Yes | No | No |
| 12 | France | No | No | No | No | No | No |
| 13 | Hainan | Yes | No | No | Yes | No | No |
| 14 | Malaysia | Yes | No | No | Yes | No | No |
| 15 | Qantas | Yes | No | No | No | No | No |
| 16 | Shandong | Yes | No | No | Yes | No | No |
| 17 | Singapore | Yes | No | No | Yes | No | No |
| 18 | Southwest | Yes | No | No | No | Yes | Yes |
| 19 | Thai | Yes | No | No | Yes | No | No |
| 20 | United | No | No | No | No | Yes | Yes |

Note: LQ represents lower quantiles (that is, quantiles 0.1 to 0.3), MQ is middle quantiles (that is, 0.4 to 0.6), and HQ is higher quantiles (that is, 0.7 to 0.9).

uncertainty-induced by the pandemic are stronger in causing the airline stocks of American, Ana, China, Delta, Hainan, Malaysia, Qantas, Shandong, Singapore, Southwest and Thai Malaysia and Thai to be volatile during bearish market condition. During the COVID-19 period, there are evidences of uncertainty induced volatility into more stocks in different market conditions. For instance, the volatility of American, Asiana, Cathay, China, China east, China south, Delta, Deutsche, Eva, Hainan, Malaysia, Shandong, Singapore and Thai are influenced during highly unfavourable market condition, the stock volatility of Cathay, Southwest and United are significantly affected when the market condition is normal. In periods of extreme market performance, pandemic uncertainty causes volatility in the stocks returns of Southwest and United airlines.

## 3.3 Summary of the results

The previous sections above have raised answers to the questions developed in this study. Firstly, uncertainty induced by health pandemics causes most of the airline stocks returns, especially at the lower quantiles, hence suggesting that pandemic induced uncertainty should be not be left out when predicting the airline stocks returns especially during market turmoil. Secondly, when predicting the volatility of the stock returns, the role of pandemic uncertainty should be considered. The findings from this study are in line with the works of [28–30], which support the effect of uncertainty in influencing the returns and volatilities of stocks at different market conditions.

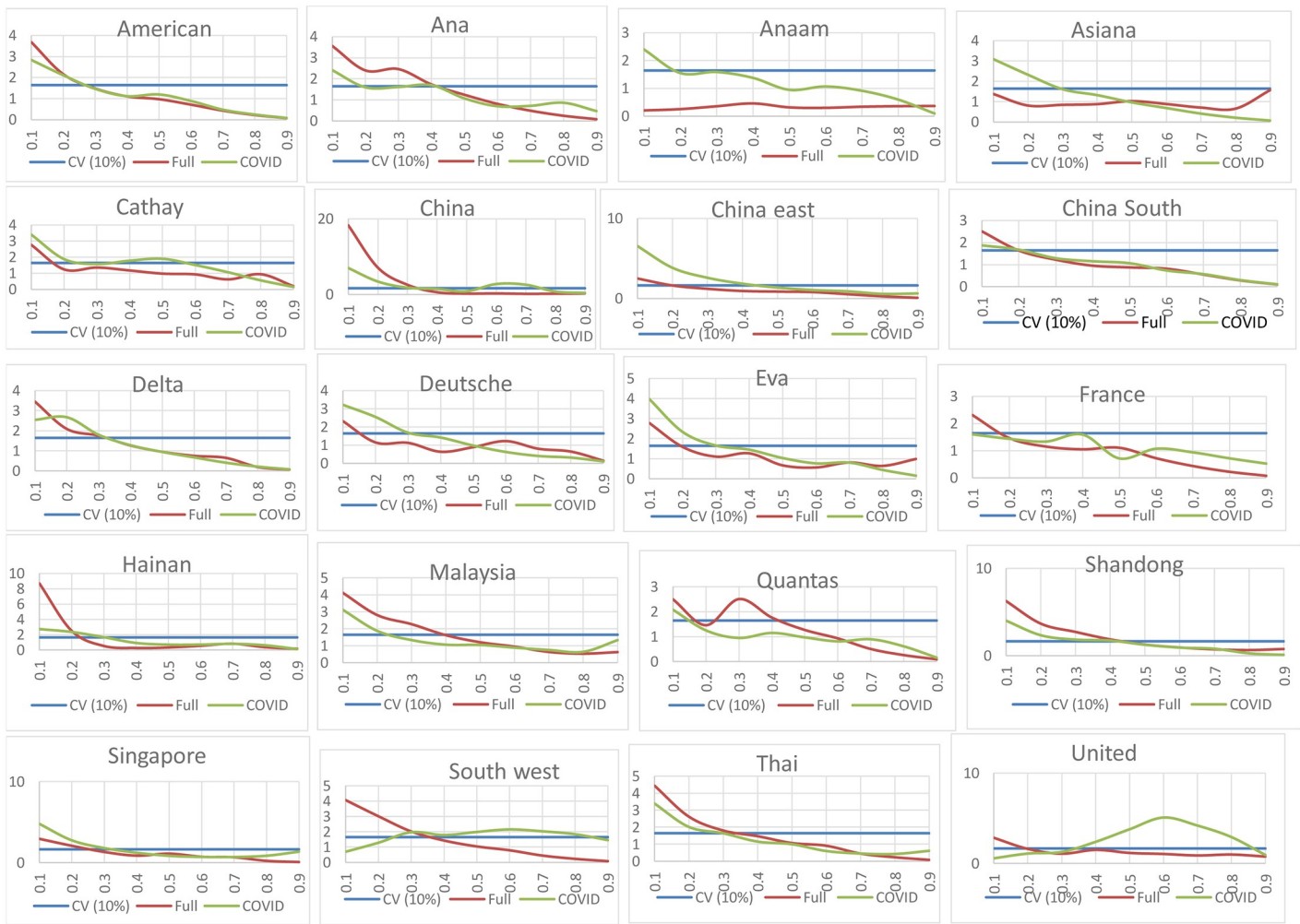

**Fig 3. Causality-in-variance result for the airlines.** Note: The figure plots the estimates of the nonparametric causality tests of the various quantiles. The y-axis report test statistics and quantiles of the airline stock returns are on x-axis. Full and COVID represents the full sample and COVID-19 pandemic sample, respectively.

## 4. Conclusion

This paper examines how uncertainty due to infectious diseases affects international airline stock returns and volatility across different quantiles. For completeness, we commenced with the standard linear Granger causality test, which, in turn, showed no evidence of uncertainty due to infectious diseases predicting airline stock returns barring American Ana and Qantas airlines. Since the results from the linear causality test cannot be relied upon due to strong evidence of nonlinearity in our data as confirmed by the BDS test, we apply the nonparametric causality-in-quantiles test, which is robust to model misspecification due to nonlinearity and regime shifts. The quantile-based nonlinear results show that, for both conditional mean and conditional variance, the disease-based uncertainty significantly affects the airline stock returns and volatility at lower and median quantiles but stronger mostly at lower quantiles. In essence, our results highlight the importance of accounting for nonlinearity when predicting airline stock returns and its volatility based on uncertainty due to infectious diseases.

As suggestions for future studies, there is still a need to delve into the drivers of airline stocks with emphasis on investors' sentiment, climate policy uncertainty and economic policy

uncertainty. Also, since our study documents that uncertainty induces volatility in the stocks, future studies should analyse how persistent the volatilities of and efficient the airline stocks are during different crises and post crises periods.

## Supporting information

**S1 Appendix.**
(DOCX)

## Author Contributions

**Conceptualization:** Ismail O. Fasanya.

**Data curation:** Johnson A Oliyide.

**Formal analysis:** Oluwasegun B. Adekoya, Johnson A Oliyide.

**Methodology:** Ismail O. Fasanya.

**Supervision:** Ismail O. Fasanya.

**Writing – original draft:** Ismail O. Fasanya, Oluwasegun B. Adekoya.

**Writing – review & editing:** Ismail O. Fasanya, Johnson A Oliyide.

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
