## [Decision Letter · Decision Letter 0]

11 Jan 2022

PONE-D-21-38713Economic uncertainty of pandemic and International airlines behaviourPLOS ONE

Dear Dr. Ismail Fasanya,

Thank you for submitting your manuscript to PLOS ONE. After careful consideration, we feel that it has merit but does not fully meet PLOS ONE’s publication criteria as it currently stands. Therefore, we invite you to submit a revised version of the manuscript that addresses the points raised during the review process.

We look forward to receiving your revised manuscript.

Kind regards,

Ricky Chia Chee Jiun

Academic Editor

PLOS ONE

Journal Requirements:

Reviewers' comments:

Reviewer's Responses to Questions

5. Review Comments to the Author

Reviewer #1: Dear/s Author/s,

Re: Manuscript "Economic uncertainty of pandemic and International airlines behavior"

Reviewer’s report:

The article deals with a very interesting topic related to the prediction of the value of shares in times as complex as the current ones derived from the COVID-19 pandemic, and very especially, for the airline sector. The article is well written, but for its publication the authors should make the following modifications:

- The study is adequately justified, but in the specific case that is exposed of the airlines, it would be advisable to have a deeper justification in relation to the variables to be studied and even add a greater number of cases, whenever possible or for subsequent studies, therefore which in this case could be exposed as future lines of research. See, in this last sense, the following article:

Bousoño-Calzón, C., Bustarviejo-Muñoz, J, Aceituno-Aceituno, P., Escudero Garzas, J.J. (2019). On the Economic Significance of Stock Market Prediction and the No Free Lunch Theorem. IEEE ACCESS. 7, 75177-75188.

- The discussion between the results and the theoretical framework from which the objective is formulated has not really taken place. These aspects are not discussed, nor are the authors exposed in the theoretical framework cited in the discussion, and with this, the contribution of the article is not clearly established.

Best regards

---

## [Author Response · Author response to Decision Letter 0]

17 Mar 2022

Re-submission of manuscript entitled: Economic uncertainty of pandemic and International airlines behavior

Manuscript ID: PONE-D-21-38713

Dear Editor, 

Thank you for reviewing our article with the above caption. We are very grateful for your constructive and valuable comments, and particularly for giving us the opportunity to make a revision. The paper has been revised to accommodate your comments.

EDITOR’s COMMENTS:

Comment 1: Please ensure that your manuscript meets PLOS ONE's style requirements, including those for file naming. The PLOS ONE style templates can be found at 

Response 1:

Many thanks for the comment. We have revised the manuscript to accommodate this.

Comment 2: Please review your reference list to ensure that it is complete and correct. If you have cited papers that have been retracted, please include the rationale for doing so in the manuscript text, or remove these references and replace them with relevant current references. Any changes to the reference list should be mentioned in the rebuttal letter that accompanies your revised manuscript. If you need to cite a retracted article, indicate the article’s retracted status in the References list and also include a citation and full reference for the retraction notice.

Response 2: Thank you for your comment. The manuscript has been adjusted to ensure that the full reference of all in-text citations are well highlighted in the reference list. Also, due to the reviewer’s comments. Some new studies were cited such as,

Adekoya, O.B., & Oliyide, J.A. (2021). How COVID-19 drives connectedness among commodity and financial markets: Evidence from TVP_VAR and causality-in-quantiles techniques. Resources Policy, 70, 101898.

Fasanya, I.O., Oliyide, J.A., Adekoya, O.B., & Agbatogun, T. (2021c). How does economic policy uncertainty connect with the dynamic spillovers between precious metals and bitcoin markets? Resources Policy, 72.

Lintner, J. (1965). The valuation of risk assets and the selection of risky investments in stock portfolios and capital budgets. The Review of Economics and Statistics 47, 13-37.

Oliyide, J.A., Adekoya, O.B., & Khan, M.A. (2021). Economic policy uncertainty and the volatility and the volatility connectedness between oil shocks and metal market: An extension. International Economics, 167, 136-150.

Sharpe, W.E. (1964). Capital-asset prices- A theory of market equilibrium under conditions of risk. Journal of Financee, 19, 425-442.

Comment 3: Thank you for stating the following financial disclosure: 

Response 3: Thank you, Editor.

Comment 4: Please clarify the sources of funding (financial or material support) for your study. List the grants or organizations that supported your study, including funding received from your institution.

Response 4: “The funders had no role in study design, data collection and analysis, decision to publish, or preparation of the manuscript.”

Comment 5: State what role the funders took in the study. If the funders had no role in your study, please state: “The funders had no role in study design, data collection and analysis, decision to publish, or preparation of the manuscript.”

Response 5: Thank you for your comment. 

Comment 6: If any authors received a salary from any of your funders, please state which authors and which funders.

Response 6:

Not applicable 

Comment 7: In your Data Availability statement, you have not specified where the minimal data set underlying the results described in your manuscript can be found. PLOS defines a study's minimal data set as the underlying data used to reach the conclusions drawn in the manuscript and any additional data required to replicate the reported study findings in their entirety. All PLOS journals require that the minimal data set be made fully available. For more information about our data policy, please see http://journals.plos.org/plosone/s/data-availability.

Response 7:

Data are available upon request from the corresponding author

REVIEWER’S COMMENTS:

Comment 1: The article deals with a very interesting topic related to the prediction of the value of shares in times as complex as the current ones derived from the COVID-19 pandemic, and very especially, for the airline sector

Response 1: Thank you for your kind words and your detailed review of our paper.

Comment 2: The study is adequately justified, but in the specific case that is exposed of the airlines, it would be advisable to have a deeper justification in relation to the variables to be studied and even add a greater number of cases, whenever possible or for subsequent studies, therefore which in this case could be exposed as future lines of research. See, in this last sense, the following article:

Bousoño-Calzón, C., Bustarviejo-Muñoz, J, Aceituno-Aceituno, P., Escudero Garzas, J.J. (2019). On the Economic Significance of Stock Market Prediction and the No Free Lunch Theorem. IEEE ACCESS. 7, 75177-75188.

Response 2: Your comment is highly appreciated. It is important mention that the selection of the airline stocks are based on data availability and on their two-letter designations from the International Air Transport Association (IATA). We have incorporated this in the data section. Also, we have suggest some critical research areas to future studies.

Comment 3: The discussion between the results and the theoretical framework from which the objective is formulated has not really taken place. These aspects are not discussed, nor are the authors exposed in the theoretical framework cited in the discussion, and with this, the contribution of the article is not clearly established.

Response 3: Thank you so much for your constructive comment. We find this comment very valuable for the re-construction of our manuscript, hence we have revisited the whole manuscript. First of all, as seen in the fourth paragraph in the introduction section, we have highlighted the gap in the literature and subsequently derive the contribution thereof. Secondly, we briefly introduce the standard capital assets pricing model. In all, we revisited the discussion of the findings where the contribution of the study are adequately linked with capital assets pricing model.

---

## [Editor Report · Decision Letter 1]

29 Mar 2022

Economic uncertainty of pandemic and International airlines behaviour

PONE-D-21-38713R1

Dear Dr. Ismail Fasanya,

We’re pleased to inform you that your manuscript has been judged scientifically suitable for publication and will be formally accepted for publication once it meets all outstanding technical requirements.

Kind regards,

Ricky Chee Jiun Chia

Academic Editor

PLOS ONE
---

## [Editor Report · Acceptance letter]

16 May 2022

PONE-D-21-38713R1 

Economic uncertainty of pandemic and International airlines behaviour  

Dear Dr. Fasanya:

I'm pleased to inform you that your manuscript has been deemed suitable for publication in PLOS ONE. Congratulations! Your manuscript is now with our production department. 

Kind regards, 

on behalf of

Dr. Ricky Chee Jiun Chia 

Academic Editor

PLOS ONE